# Low-intensity pulsed ultrasound therapy suppresses coronary adventitial inflammatory changes and hyperconstricting responses after coronary stent implantation in pigs in vivo

Tasuku Watanabe[1], Yasuharu Matsumoto[1], Kensuke Nishimiya[1], Tomohiko Shindo[1], Hirokazu Amamizu[1], Jun Sugisawa[1], Satoshi Tsuchiya[1], Koichi Sato[1], Susumu Morosawa[1], Kazuma Ohyama[1], Tomomi Watanabe-Asaka[2], Moyuru Hayashi[2], Yoshiko Kawai[2], Jun Takahashi[1], Satoshi Yasuda[1], Hiroaki Shimokawa[1,3]*

1 Department of Cardiovascular Medicine, Tohoku University Graduate School of Medicine, Sendai, Japan, 2 Division of Physiology, Tohoku Medical and Pharmaceutical University, Sendai, Japan, 3 International University of Health and Welfare, Narita, Japan

* shimo@cardio.med.tohoku.ac.jp

## Abstract

### Backgrounds

We demonstrated that coronary adventitial inflammation plays important roles in the pathogenesis of drug-eluting stent (DES)-induced coronary hyperconstricting responses in pigs in vivo. However, no therapy is yet available to treat coronary adventitial inflammation. We thus developed the low-intensity pulsed ultrasound (LIPUS) therapy that ameliorates myocardial ischemia by enhancing angiogenesis.

### Aims

We aimed to examine whether our LIPUS therapy suppresses DES-induced coronary hyperconstricting responses in pigs in vivo, and if so, what mechanisms are involved.

### Methods

Sixteen normal male pigs were randomly assigned to the LIPUS or the sham therapy groups after DES implantation into the left anterior descending (LAD) coronary artery. In the LIPUS group, LIPUS (32 cycles, 193 mW/cm$^2$) was applied to the heart at 3 different levels (segments proximal and distal to the stent edges and middle of the stent) for 20 min at each level for every other day for 2 weeks. The sham therapy group was treated in the same manner but without LIPUS. At 4 weeks after stent implantation, we performed coronary angiography, followed by immunohistological analysis.

### Results

Coronary vasoconstricting responses to serotonin in LAD at DES edges were significantly suppressed in the LIPUS group compared with the sham group. Furthermore, lymph

**Data Availability Statement:** All relevant data are within the manuscript and its Supporting Information files.

**Funding:** This work was supported in part by the grants-in-aid for the Scientific Research (18K15877, 19K11762, 19K17511), Mitsui Sumitomo Insurance Welfare Foundation, and Sakakibara Memorial Research Grant from the Japan Research Promotion Society for Cardiovascular Diseases. The funders had no role in study design, data collection and analysis, decision to publish, or preparation of the manuscript. The authors received no specific funding for this work.

**Competing interests:** The authors have declared that no competing interests exist.

transport speed in vivo was significantly faster in the LIPUS group than in the sham group. Histological analysis at DES edges showed that inflammatory changes and Rho-kinase activity were significantly suppressed in the LIPUS group, associated with eNOS up-regulation and enhanced lymph-angiogenesis.

## Conclusions

These results suggest that our non-invasive LIPUS therapy is useful to treat coronary functional abnormalities caused by coronary adventitial inflammation, indicating its potential for the novel and safe therapeutic approach of coronary artery disease.

## Introduction

Although endothelial dysfunction and subsequent intimal thickening are thought to be an initial step for coronary artery disease (CAD) [1], it has been suggested that coronary adventitial inflammation also play pivotal roles in the pathogenesis of the disorder [2]. Furthermore, coronary adventitial inflammation plays an important role in the pathogenesis of acute coronary syndrome (ACS) [3]. We previously demonstrated that coronary adventitial inflammation, including vasa vasorum formation, inflammatory cells migration, and cytokines secretion by perivascular adipose tissue (PVAT), is involved in coronary hyperconstricting responses after drug-eluting stents (DES) implantation in pigs and patients with vasospastic angina (VSA) [4–8]. Moreover, we recently demonstrated that impairment of cardiac lymphatic drainage function exacerbates adventitial inflammation, and medial vascular smooth muscle cell (VSMC) hyperconstriction through activation of Rho-kinase (a central molecular switch of coronary spasm) in pigs in vivo [9]. However, currently, no therapeutic approach that specifically targets on coronary adventitial inflammation is available.

We have recently developed a non-invasive therapy with low-intensity pulsed ultrasound (LIPUS) that exerts therapeutic angiogenesis [10–14]. We demonstrated that the angiogenic effects of LIPUS is mainly mediated by endothelial nitric oxide synthase (eNOS) upregulation, ameliorating left ventricular dysfunction in animal models of acute and chronic myocardial ischemia with no adverse effects [10, 11]. Furthermore, the LIPUS therapy also exerts anti-inflammatory effects [12, 15]. However, it remains to be examined whether the LIPUS therapy is able to suppress coronary adventitial inflammation.

In the present study, we thus examined whether our LIPUS therapy suppresses coronary hyperconstricting responses in pigs after DES implantation in vivo, and if so, what mechanisms are involved.

## Methods

### Study protocol

All animal care and experimental studies were performed in accordance with the Guide for the Care and Use of Laboratory Animals published by the U.S. National Institute of Health (NIH Publication, 8th Edition, 2011) and ARRIVE guidelines, and were approved by the Institutional Committee for Use of Laboratory Animal of Tohoku University (2017MdA-139,278–01).

## Animal preparation

Sixteen male Yorkshire pigs (3–4 month-old, male neutered, weighing 35–45 kg) were pre-treated orally with aspirin (300 mg/day) and clopidogrel (150 mg/day) for 2 days before stent implantation (**Fig 1A**). After sedation with medetomidine [0.1 mg/kg, intramuscular injection (IM)] and midazolam (0.2 mg/kg, IM) followed by inhaled sevoflurane (2–5%) and heparinization [5,000 U, intravenous injection (IV)], animals were implanted everolimus-eluting stents (EES, Promus$^{TM}$, Boston Scientific Corporation, MA, USA) in their left anterior descending coronary artery (**S1 Table**). Balloon inflation ratio was adjusted to achieve an overstretch ratio of 1.0–1.1 under the guidance of intravascular ultrasound (ViewIT$^{®}$, Terumo Corporation, Tokyo, Japan) (**S1 Table**). Animals were alternately allocated to either the LIPUS or the sham group (n = 8 each). Anti-platelet therapy with oral aspirin (100 mg/day) and clopidogrel (75 mg/day) was continued until euthanasia. At 4 weeks after DES implantation, we performed coronary angiography (CAG) to evaluate coronary vasomotion at the proximal and distal stent edge segments in vivo [4–7, 9, 16]. Median thoracotomy was then performed to examine cardiac lymph transport function [9]. Animals were finally euthanized with a lethal dose of potassium chloride (0.25 mEq/kg, IV) under deep anesthesia with inhaled 5% sevoflurane, and the hearts were dissected for ex vivo experiments.

## LIPUS therapy

An ultrasound device (Prosound α10; HITACHI, Ltd., Tokyo, Japan) was chosen for the LIPUS therapy, which irradiation settings could be modified at ease. Based on our previous studies [10, 11], the LIPUS therapy was performed under the following settings; frequency = 1.875 MHz, pulse repetition frequency = 4.90 kHz, number of cycles = 32, voltage applied to each transducer element = 17.67–22.38 volts, and spatial peak temporal average intensity (Ispta) = 117–174 mW/cm$^2$. The power of LIPUS was estimated 0.25 W/cm$^2$, and the beams were irradiated from the sector-shaped probe and were focused at 6 cm depth [10, 11]. The number of cycles of pulsed ultrasound represents that of acoustic waves per 1 pulse, while 1 cycle is used for diagnostic ultrasound devices (**S1A** and **S1B Fig**). The voltage applied to each transducer element was controlled to keep estimated Ispta of LIPUS below the upper limit of acoustic output standards (<720 mW/cm$^2$) for diagnostic ultrasound devices (US Food and Drug Administration's Track 3 Limits) and to prevent the ultrasound probe from temperature rise [10, 11].

Pigs were placed left-side-up under 2–5% inhaled sevoflurane to receive the LIPUS therapy [10]. LIPUS was percutaneously irradiated to the 3 different sites around DES in LAD, including the sites proximal and distal to the stents, middle portions of the stents (**Fig 1B**). Each site was determined by co-registering radiopaque echo probe with the stent architecture (**S1C Fig**). A LIPUS session was performed for 20 min x 3 sites per day, and was repeated every other day for 2 weeks beginning immediately after DES implantation (**Fig 1A**) as previously described [10]. The non-treated sham group was also received anesthesia and simply placed echo probe toward the 3 different sites under fluoroscopy.

## In vivo assessment of coronary vasomotor responses

At 4 weeks after DES implantation, we examined coronary vasomotor responses to serotonin [10 and 100 µg/kg, intracoronary injection (IC)] before and after hydroxyfasudil (30 and 300 µg/kg, IC for 3 min), a specific Rho-kinase inhibitor (Asahi Kasei Pharma, Tokyo, Japan) [16], nitroglycerin (10 µg/kg, IC), and bradykinin (0.1 µg/kg, IC) before and after N$^G$-mono-methyl-L-arginine (1 mg/kg, IC for 10 min) [4–7, 9, 16]. Quantitative coronary angiography (QCA) was performed at the proximal and distal stent edge segments 5 mm apart from the

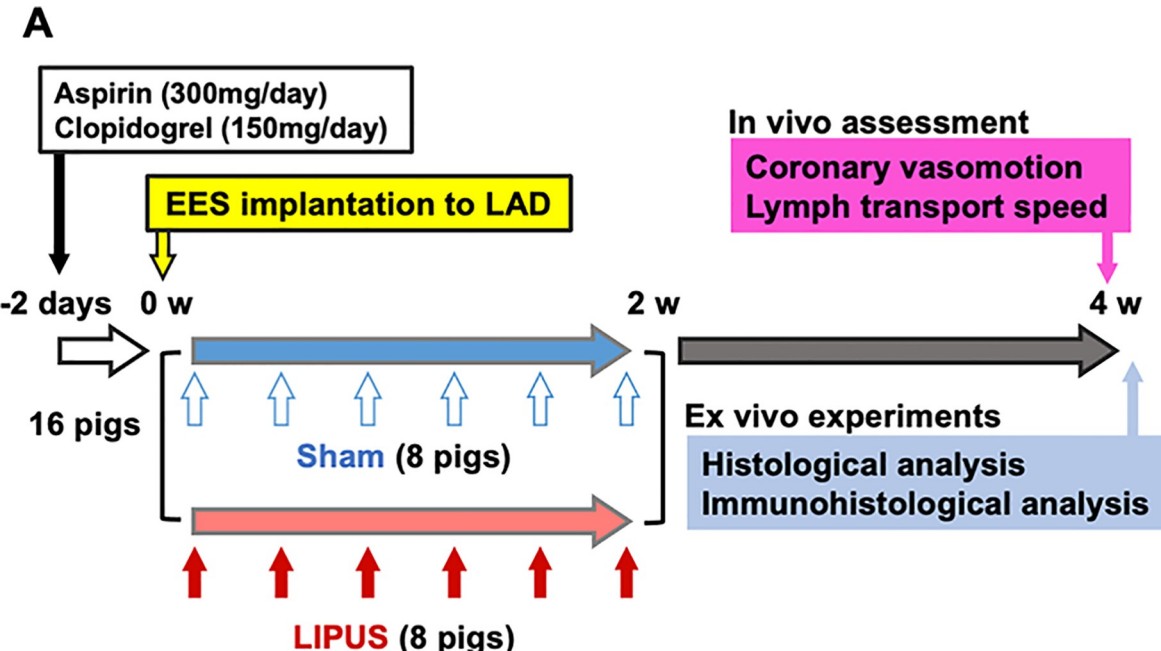

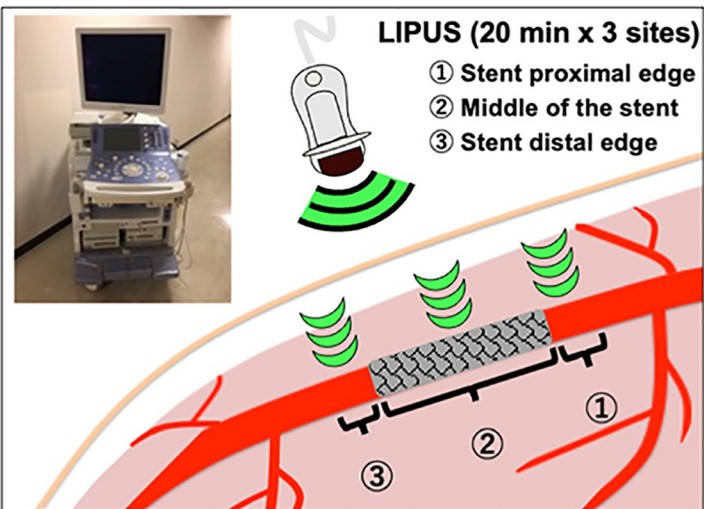

**Fig 1. Study protocol.** (**A**) Sixteen pigs were implanted EES in the LAD, and were allocated to either the sham or the LIPUS group. Animals were then anesthetized to receive the sham or the LIPUS treatment (every other day for 2 weeks). At 4 weeks after EES implantation, coronary vasomotion and cardiac lymph transport speed were assessed in vivo. After euthanization, the heart was dissected for ex vivo histological analysis. (**B**) LIPUS was irradiated to the sites proximal and distal to the stent edges, and the middle portions of EES (20 min x 3 sites/day). CAG = coronary angiography; EES = everolimus-eluting stent; LAD = left anterior descending coronary artery; LIPUS = low-intensity pulsed ultrasound.

stent as previously described [4–7, 9]. The segments 10 mm distal to the ostium of the left circumflex coronary arteries (LCx) were used for evaluating coronary vasomotion of the untreated vessels.

### In vivo evaluation of cardiac lymph transport speed

After median thoracotomy, cardiac lymph transport speed was evaluated by intramuscularly injecting 1 mg ICG (Diagnogreen 25 mg, Daiichi Sankyo Co. Ltd., Tokyo, Japan) to the left ventricular apex. ICG's behavior from apex to base on the beating heart was monitored with the photodynamic eye near-infrared camera system (PDE, Hamamatsu Photonics, Shizuoka, Japan) [9, 17]. Cardiac lymph transport speed was calculated by the following formula; [the total length of the path of ICG (mm)/time constant (min)] [9].

### Histological analysis

After the removal of the hearts, the left coronary arteries were perfused with 10% neutral buffered formalin via a constant perfusion pressure system (120 cm $H_2O$). Samples 5 mm apart from the stent were isolated and cut into a 3 μm-thick slice [4–7, 9, 16]. All sections were stained with hematoxylin-eosin and Masson's trichrome (MT), and digitized with Axio-Vision® Software (Release 4.5, Zeiss, Jena, Germany). Coronary cross-sections in MT were manually segmented for histomorphometry using ImageJ (U.S. National Institute of Health, Maryland, USA). Adventitial area was calculated by the following formula; [area outside the external elastic lamina (EEL) within a distance of the thickness of neointima plus media—EEL area] [4–7]. Individual adipocytes were selected in 3 random high-power fields, and adipocyte size was expressed as an average of the perpendicular maximum and minimum axes [7, 9].

### Immunohistological analysis

Details are available in **Supplementary material online.** Briefly, we used antibodies of eNOS, LYVE-1 for lymphatic vessels [9], von Willebrand factor (vWF) for vasa vasorum, VEGF-A/-C, VEGFR2/3, CD68 for macrophages, and IL-1β (**S2 Table**). Using ImageJ, luminal structures positive for eNOS, LYVE-1, vWF, VEGF-A/-C, and VEGFR2/3 antibodies were counted and divided by the adventitial area (**S3 Table**). Adventitial CD68 and IL-1β-positive cells were counted in the 3 randomly selected high-power fields [4, 7, 9]. The adiponectin distribution to PVAT was evaluated by the following formula; [area positive for immunostainings/total area] at 100x magnification [7, 9]. The extent of ROCK1/2 and activation pMYPT-1 in medial VSMC was semi-quantitatively evaluated at 12 radial subparts using the following scale; 0 = none, 1 = slight, 2 = moderate, and 3 = high, as previously described [16].

### Statistical analysis

Results are expressed as mean±standard error of mean. Comparison of the QCA and histomorphometry was performed by unpaired, 2-sided Welch's $t$-test. Comparison of the semi-quantitative analysis was performed by using Mann-Whitney $U$-test. Box-and-whisker plots express that the central box covers the interquartile range, with the median indicated by the line within the box. Outliers are plotted outside the 1.5 interquartile ranges. Statistical analysis was performed with R version 3.6.1 (R Foundation for Statistical Computing, Vienna, Austria). A value of P<0.05 was considered to be statistically significant.

## Results

### LIPUS therapy ameliorates DES-induced coronary hyperconstricting responses in vivo

CAG at 4 weeks after DES implantation to the left anterior descending coronary arteries (LAD) showed no in-stent restenosis (**Fig 2A and 2D**). Importantly, coronary

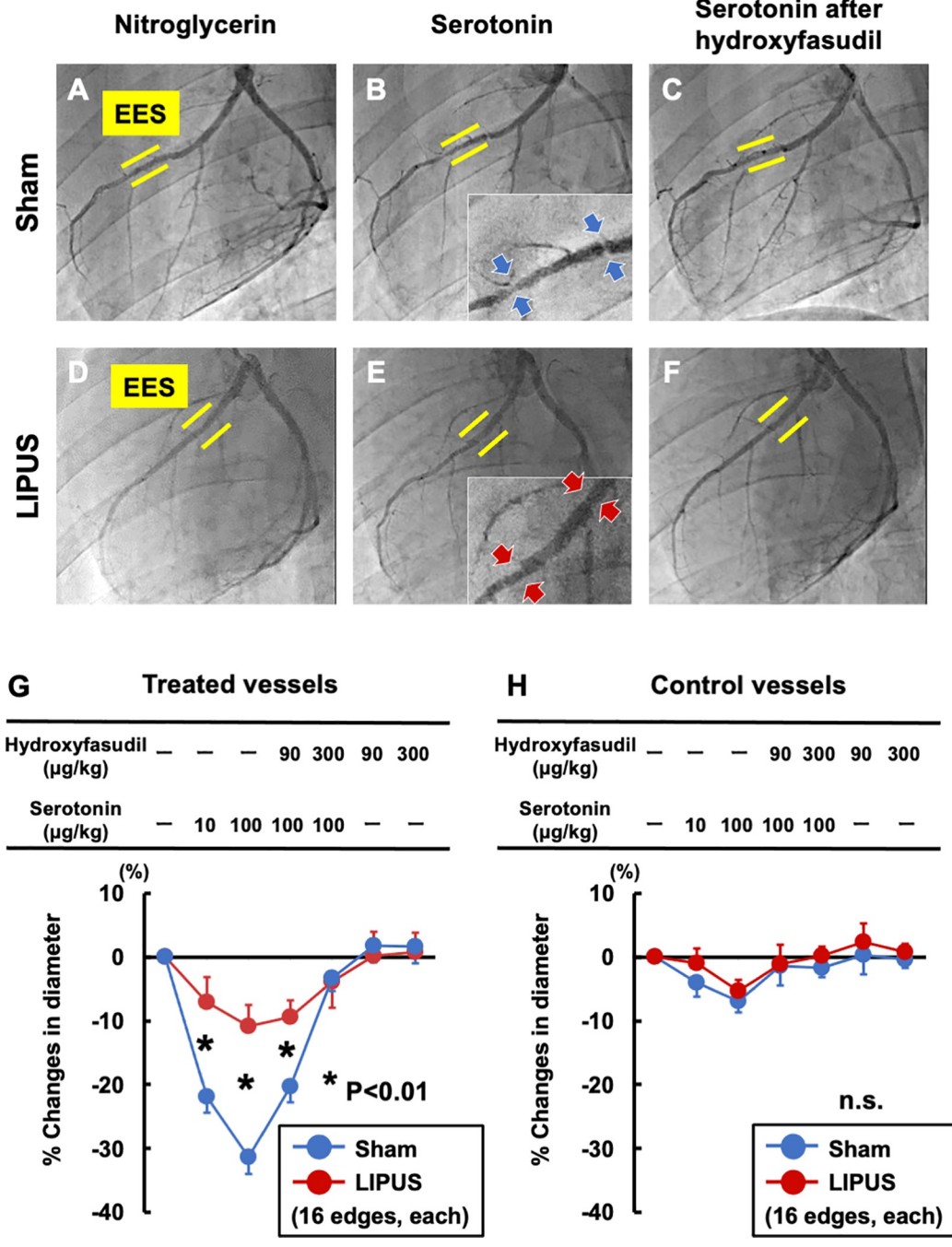

**Fig 2. Coronary vasoconstricting responses to serotonin before and after hydroxyfasudil.** Representative CAG at 4 weeks (**A**, **D**) after intracoronary nitroglycerin (10 μg/kg), (**B**, **E**) serotonin (100 μg/kg), and (**C**, **F**) serotonin after pre-treatment with hydroxyfasudil (a selective Rho-kinase inhibitor, 300 μg/kg). Coronary hyperconstricting responses at the stent edges (blue and red arrows) of EES (yellow lines) were markedly suppressed in the LIPUS group compared with the sham group (**B**, **E**), which were fully abolished by hydroxyfasudil (**C**, **F**). (**G**) Coronary vasoconstriction to serotonin was significantly attenuated in the LIPUS group compared with the sham group, which was prevented by hydroxyfasudil. (**H**) Vasoconstriction was not evident in the control arteries. Results are expressed as mean±SEM. SEM = standard error of mean; other abbreviations as in **Fig 1**.

hyperconstricting responses to serotonin were markedly suppressed in the LIPUS group compared with the sham group (serotonin 100 μg/kg; P<0.001, **Fig 2B and 2E**). Pre-treatment with hydroxyfasudil, a selective Rho-kinase inhibitor [8, 16], abolished serotonin-induced coronary hyperconstricting responses (**Fig 2C and 2F**). QCA analysis showed that coronary vasoconstriction to serotonin was significantly attenuated in the LIPUS group compared with the sham group, and was fully prevented by hydroxyfasudil (**Fig 2G**). In contrast, coronary vasoconstriction was not evident in the untreated control vessels (**Fig 2H**). Endothelium-dependent and -independent vasodilating responses were comparable in the 2 groups (**S4 Table**). Inter- and intra-observer variabilities for QCA were within acceptable ranges (**S2 Fig**).

## LIPUS improves cardiac lymphatic vessel function in vivo

Intriguingly, near-infrared camera showed that indocyanine green (ICG) injected to the apex moved up faster on the LIPUS-treated heart compared with the sham (**Fig 3A–3F**). Quantitatively, lymph transport speed was significantly faster in the LIPUS compared with the sham group (P = 0.015, **Fig 3G**). Lymph transport speed was negatively correlated with the extent of coronary vasoconstriction to serotonin (R = 0.76, **Fig 3H**). The LIPUS therapy did not alter the number of lymphatic vessels in the LAD coronary segments at 20 mm distal to the stent, which were not irradiated LIPUS (**S3 Fig**).

## LIPUS augments cardiac lymphatic vessels but not blood vessels

At the coronary segments of 5 mm proximal or distal to the stent edges, eNOS-positive luminal structures were significantly increased in the LIPUS compared with the sham group (P = 0.002, **Fig 4A, 4E and 4I**). Importantly, lymphatic vessel endothelial hyaluronan receptor-1 (LYVE1)-positive lymphatic vessels and lymphangiogenic markers of vascular endothelial growth factor (VEGF)-C/VEGF receptor-3-positive cells were all prominent in the LIPUS group (LYVE-1, P<0.001; VEGF-C, P = 0.025; VEGFR3, P = 0.028; **Fig 4B–4D, 4F–4H and 4J–4L**). A significant positive correlation between eNOS expression and lymphatic vessels indicated their co-localization (R = 0.56, **Fig 4M**), both of which were negatively correlated with the extent of coronary vasoconstriction to serotonin (eNOS, R = 0.55; LYVE1, R = 0.46, **Fig 4N and 4O**). Furthermore, there was a strong positive correlation between the number of lymphatic vessels and lymph transport speed (R = 0.53, **Fig 4P**). In contrast, adventitial vasa vasorum formation and related-angiogenic factors (VEGF-A/VEGFR2) were comparable between the 2 groups, and vasa vasorum showed no association with eNOS expression (vasa vasorum, P = 0.212; VEGF-A, P = 0.640; VEGFR2, P = 0.623, **S4 Fig**). These results suggest that the angiogenic effect of LIPUS was limited to lymphatic vessels but not to blood vessels. Lymphatic vessels were significantly thinner than vasa vasorum, and thinner lymphatic vessels showed more eNOS expression (R = 0.52, **S5 Fig**).

## LIPUS suppresses adventitial and PVAT inflammatory changes

CD68-positive macrophage infiltration and IL-1β expression were significantly suppressed in the LIPUS group compared with the sham group (CD68, P<0.001; IL-1β, P<0.001, **Fig 5A–5H**). Furthermore, significant positive correlations were noted between the number of CD68-positive macrophages/IL-1β-positive cells and the extent of coronary vasoconstriction to serotonin (CD68, R = 0.57; IL-1β, R = 0.56, **Fig 5I and 5J**). In contrast, lymphatic vessels were negatively correlated with these inflammatory markers (CD68, R = 0.41; IL-1β, R = 0.41, **Fig 5K and 5L**). For PVAT, enlarged adipocytes with loss of anti-inflammatory adiponectin secretion were noted in the sham group but not in the LIPUS group (adipocyte, P<0.001; adiponectin, P<0.001, **S6 Fig**).

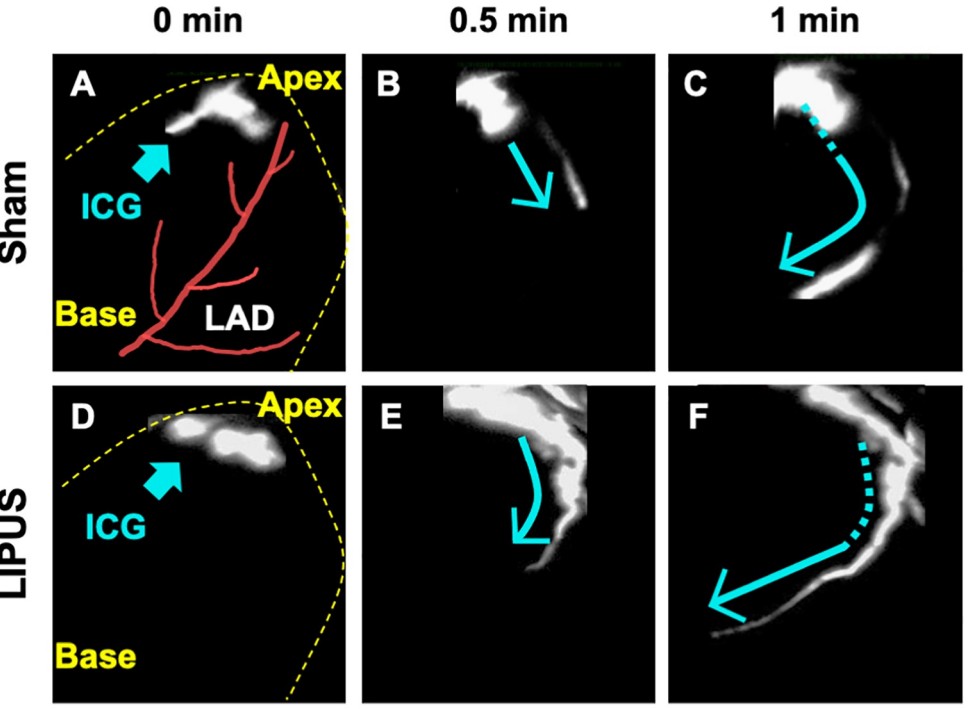

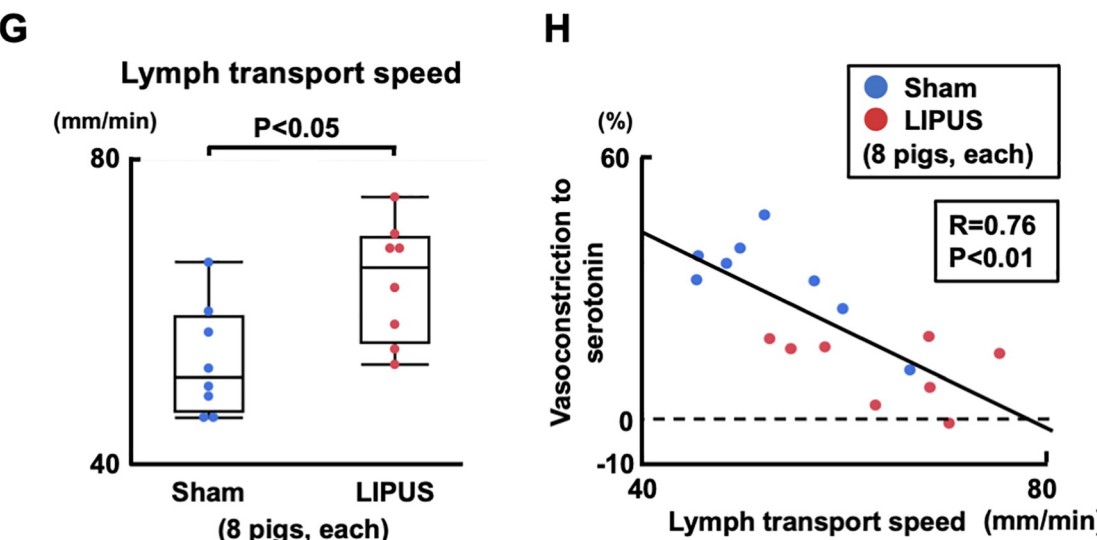

**Fig 3. In vivo assessment of lymph transport speed.** (**A-F**) Serial still images of near-infrared camera with ICG injected to beating-hearts at 4 weeks. ICG was promptly transported from the apex to the base of the LIPUS-treated heart compared with the sham heart. (**G**) Lymph transport speed was significantly faster in the LIPUS group compared with the sham group, (**H**) which was negatively correlated with coronary vasoconstriction to serotonin (100 μg/kg). Results are expressed as mean±SEM. ICG = indocyanine green; other abbreviations as in **Figs 1** and **2**.

## LIPUS suppresses Rho-kinase expressions and activation in coronary VSMC

In the medial VSMC layers of the coronary artery, Rho-kinase expressions [Rho-associated protein kinase (ROCK)1,2] and activation [phosphorylated myosin phosphatase (pMYPT)-1]

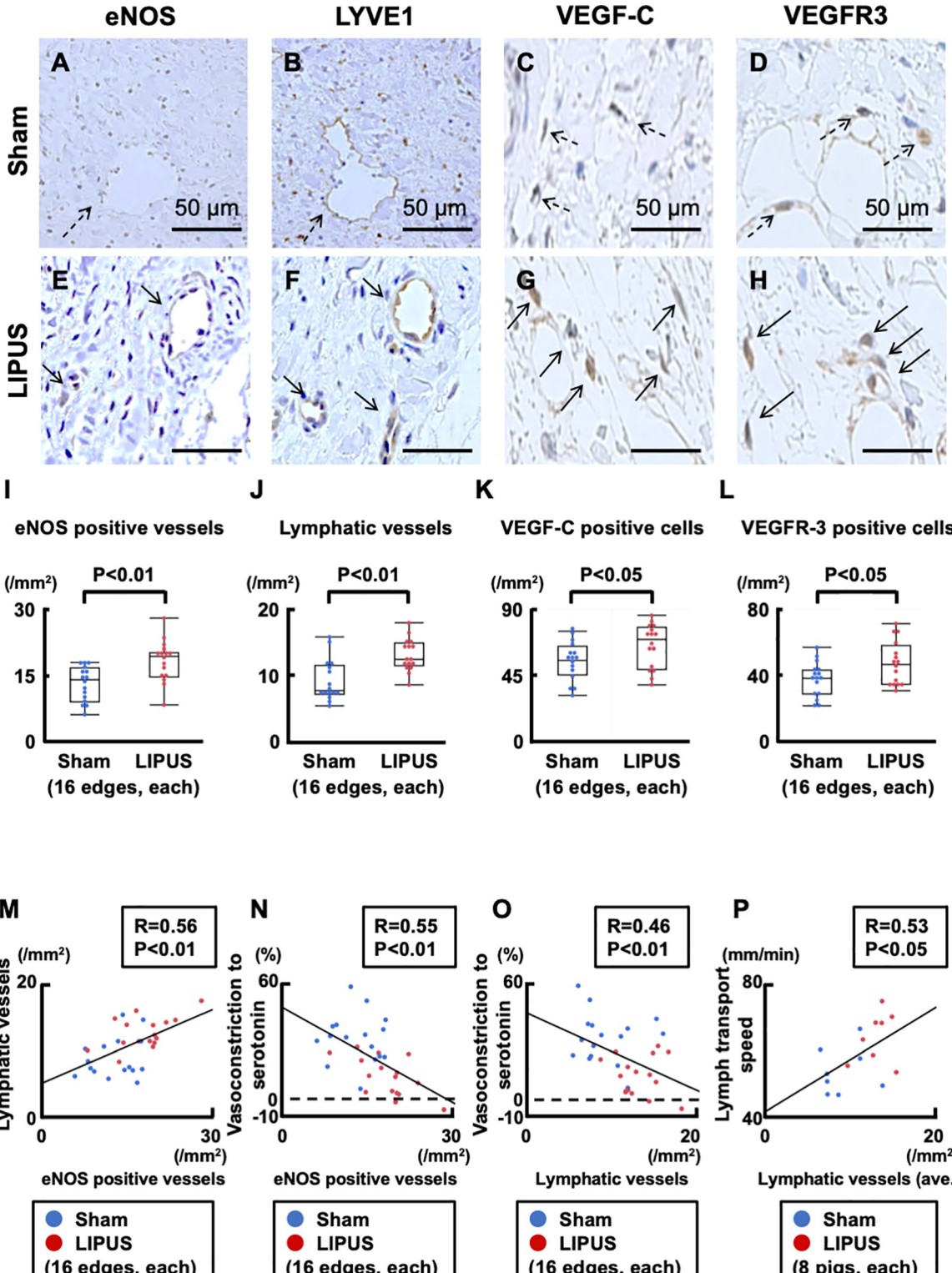

**Fig 4. Immunohistochemistry for cardiac lymph-angiogenesis.** (**A**, **E**) Representative immunohistology of eNOS-positive luminal structures, (**B, F**) LYVE-1 positive lymphatic vessels (**B, F**), VEGF-C (**C, G**), and VEGFR3 (**D, H**) in the adventitia of vasoconstricting portions. As denoted by arrows in **A-H**, (**I**) cellular structures positive for eNOS, (**J**) LYVE-1, (**K**) VEGF-C and (**L**) VEGFR3 were significantly increased in the LIPUS group compared with the sham group. (**M**) A significant positive correlation between eNOS-positive vessels vs. lymphatic vessels. (**N, O**) Significant negative correlations between eNOS-positive/lymphatic vessels vs. vasoconstriction to

serotonin (100 μg/kg). (**P**) A significant positive correlation between lymphatic vessels vs. lymph transport speed. Results are expressed as mean±SEM. eNOS = endothelial nitric oxide synthase; LYVE-1 = lymphatic vessel endothelial hyaluronan receptor-1; VEGF = vascular endothelial growth factor; VEGFR = vascular endothelial growth factor receptor; other abbreviations as in **Figs 1** and **2**.

were all significantly suppressed in the LIPUS compared with the sham group (ROCK1, P = 0.003; ROCK2, P<0.001; pMYPT1, P = 0.010, **Fig 6A–6I**). The extent of Rho-kinase expressions/activation was positively correlated with that of coronary vasoconstriction to serotonin (ROCK1, R = 0.49; ROCK2, R = 0.62; pMYPT1, R = 0.61, **Fig 6J–6L**).

## Discussion

The major findings of the present study were that (1) the LIPUS therapy suppressed DES-induced coronary hyperconstricting responses in pigs in vivo, (2) the LIPUS therapy enhanced cardiac lymphatic vessel formation associated with eNOS up-regulation, with no effects on adventitial vasa vasorum, (3) the LIPUS therapy improved lymph transport function abrogated adventitial inflammatory changes, and (4) the LIPUS therapy significantly suppressed Rho-kinase activation at the vasoconstricting coronary segment (**Fig 7**). To the best of our knowledge, this is the first study demonstrating that the LIPUS therapy exerts beneficial effects on DES-induced coronary adventitial inflammation and subsequent coronary hyperconstricting responses in vivo.

### Angiogenic and lymph-angiogenic effects of the LIPUS therapy

In order to establish a non-invasive therapy for CAD, we developed the LIPUS therapy that exerts angiogenic effect on ischemic area in pigs and mice in vivo [10, 11]. In addition to its angiogenic effect, the present study demonstrates for the first time that LIPUS also exerts lymph-angiogenic effects with subsequent inhibitory effects on coronary adventitial inflammatory changes.

The LIPUS therapy significantly increased lymphatic vessels compared with the sham group. eNOS-positive cells were prominent in the adventitia of the LIPUS-treated group. It has been reported that eNOS regulates lymphatic function by decreasing micro-lymphatic resistance [18] and that eNOS exerts anti-inflammatory effects by itself [19]. In the present study, lymph-angiogenesis markers (VEGF-C/VEGFR-3) were significantly enhanced by the LIPUS therapy. Owing to mechanotransduction induced by LIPUS [11–14], stretch of lymphatic endothelial cells may stimulate VEGF-C/VEGFR3 expressions [20]. These mechanisms may be involved in the improved lymphatic functions in the LIPUS group.

Contrary to our previous studies with other ischemia models [10, 11], LIPUS had no angiogenic effect on vasa vasorum formation in the present model with DES implantation. It is possible that LIPUS exerts angiogenic effects in ischemic tissue but none in non-ischemic tissue. In fact, in our mouse model of acute myocardial infarction, the LIPUS therapy induced angiogenesis to a greater extent at peri-infarct area than normal area [11]. Furthermore, in the present study, wall thickness was significantly thinner in lymphatic vessels than in vasa vasorum, and was negatively correlated with eNOS expression. Vasa vasorum is exposed to constant dynamic blood flow and altering shear stress, whereas lymphatic flow is rather static [21]. Thus, it is conceivable that the LIPUS therapy may induce drastic change in shear stress in thin-walled lymphatic vessels than in blood vessels. Such altered shear stress might cause eNOS up-regulation [22] predominantly in lymphatic vessels. Indeed, supporting this notion, we previously demonstrated that LIPUS-induced angiogenesis was evident in thin-walled microcapillaries in pigs with chronic myocardial ischemia [10, 11].

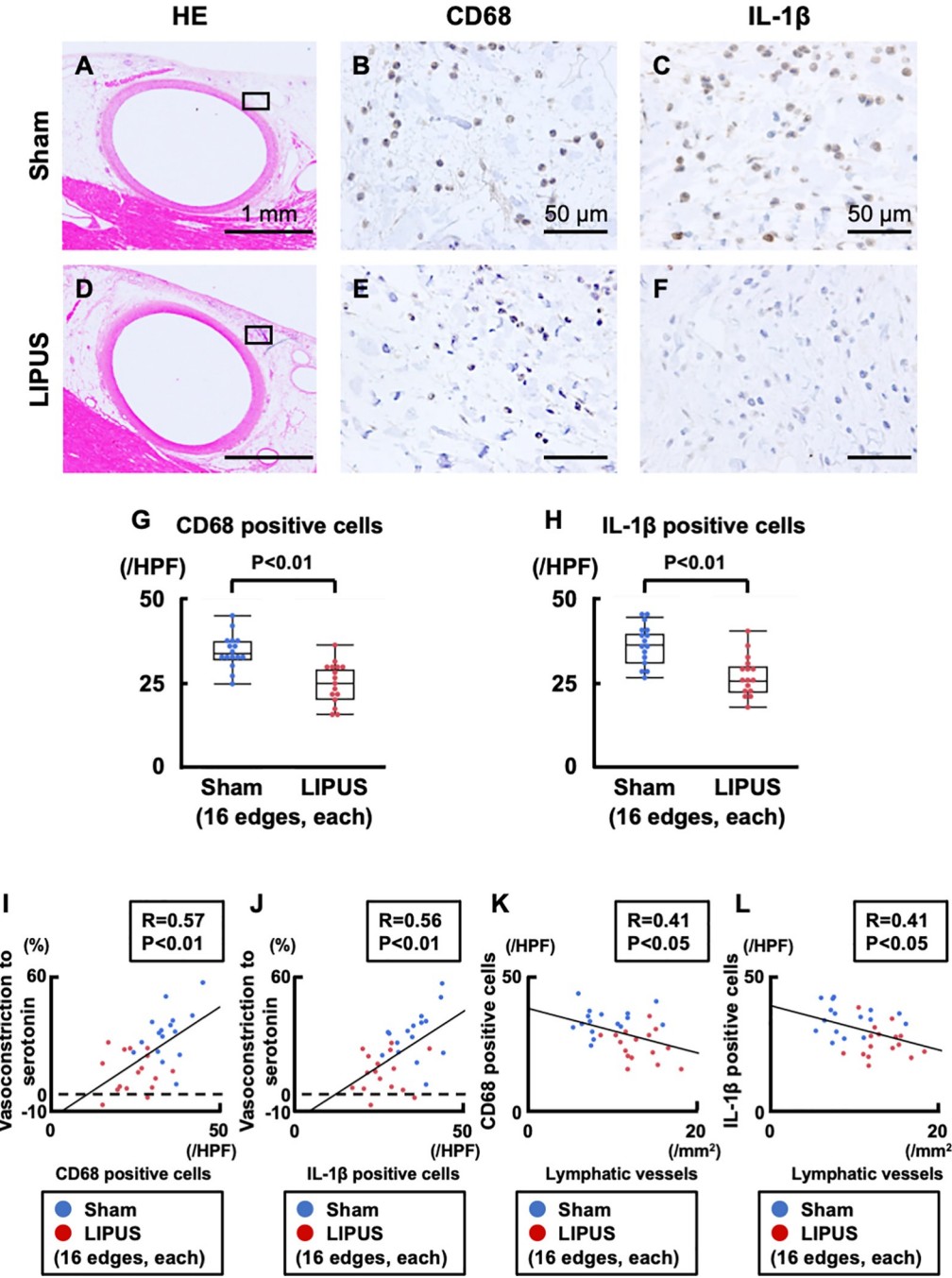

**Fig 5. Immunohistochemistry for adventitial inflammatory changes.** (**A, D**) Low-magnified HE stainings, and representative immunohistology of (**B, E**) CD68-positive macrophages and (**C, F**) IL-1β-positive cells in the adventitia of vasoconstricting portions. (**G, H**) Macrophages and IL-1β-positive cells were significantly decreased in the LIPUS group compared with the sham group. (**I, J**) Significant negative correlations between lymphatic vessels vs. CD68-/IL-1β-positive cells. (**K, L**) Significant positive correlations between CD-68/IL-1β-positive cells vs. vasoconstriction to serotonin (100 μg/kg). Results are expressed as mean±SEM. HE = hematoxylin and eosin; IL-1β = interleukin-1β; other abbreviations as in **Figs 1** and **2**.

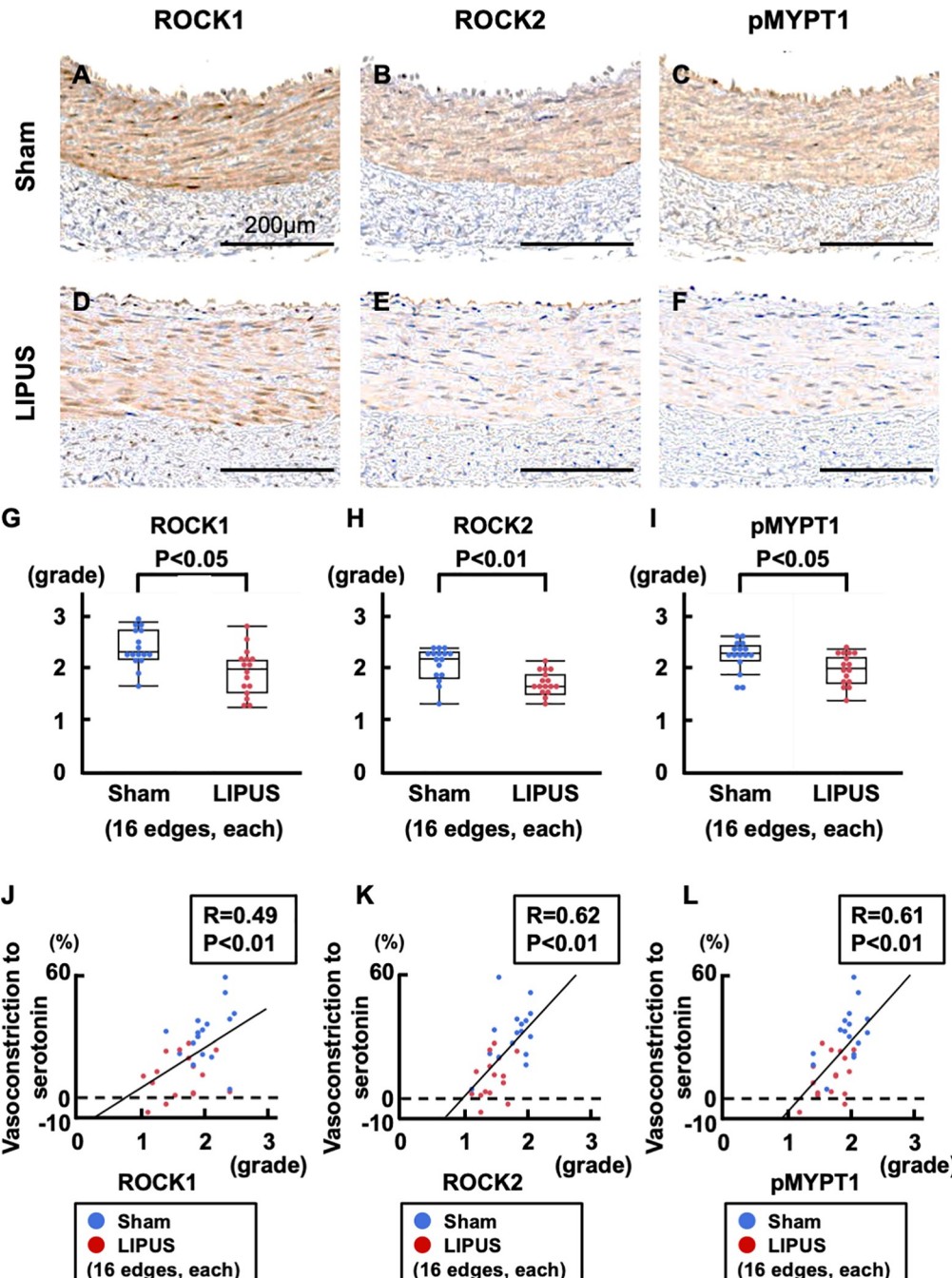

**Fig 6. Immunohistochemistry for Rho-kinase expressions and activation in the media.** (**A**, **D**) Representative immunohistology of Rho-kinase expressions for ROCK1, (**B**, **E**) and ROCK2, and (**C**, **F**) Rho-kinase activation as pMYPT1 in the medial layers of vasoconstricting portions. (**G-I**) Immunoreactivities for ROCK1/2 and pMYPT1 were all attenuated in the LIPUS group as compared with the sham group. (**J-L**) Significant correlations between ROCK1/ROCK2/pMYPT1 vs. vasoconstriction to serotonin (100 μg/kg). Results are expressed as mean±SEM.
pMYPT1 = phosphorylated myosin phosphatase target subunit-1; ROCK1 = Rho-kinase β; ROCK2 = Rho-kinase α; other abbreviations as in **Figs 1** and **2**.

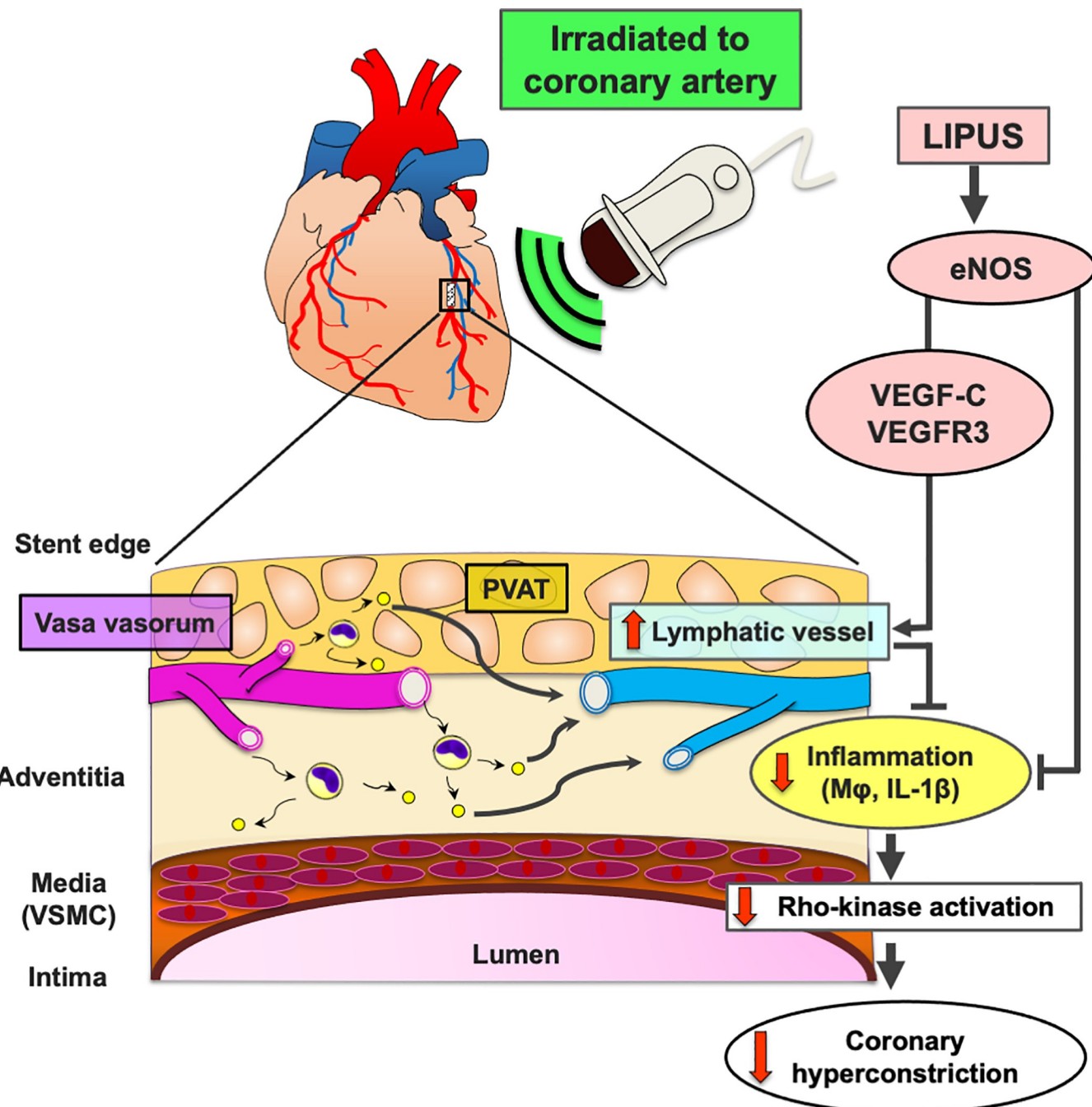

**Fig 7. LIPUS ameliorates DES-induced coronary adventitial inflammation and resultant coronary hyperconstricting responses in pigs in vivo.** DES implantation induces coronary adventitial and PVAT inflammatory changes, including adventitial vasa vasorum augmentation, Mφ infiltration, and cytokine expressions in pigs in vivo. Importantly, cardiac lymphatic vessels act as a drainage for inflammatory changes, and thus impaired lymphatic function exacerbates adventitial inflammation, and subsequent medial VSMC hypercontraction through Rho-kinase activation (a central molecular switch of coronary artery spasm). In the present study, we were able to demonstrate that the LIPUS therapy enhanced coronary lymph-angiogenesis through up-regulations of eNOS and VEGF-C/VEGFR3, which prompted lymph transport speed for inflammatory cells (Mφ) and inflammatory cytokines (IL-1β). Mechanistically, the LIPUS therapy suppressed Rho-kinase activation and subsequent medial VSMC hypercontraction. DES = drug-eluting stent; Mφ = macrophage; PVAT = perivascular adipose tissue; VSMC = vascular smooth muscle cell; other abbreviations as in **Figs 1–4**.

## Important roles of cardiac lymphatic vessels in anti-inflammatory effects of LIPUS

In the LIPUS group, improved cardiac lymphatic function was associated with attenuated adventitial inflammation. The extent of cardiac lymph-angiogenesis was positively correlated with lymph transport speed, and negatively with adventitial inflammatory changes. These results indicate that micron-level lymph-angiogensis surrounding coronary arteries ameliorates the cardiac lymphatic vessel function in the LIPUS group. Previous studies demonstrated that lymph-angiogenesis contributes to improving inflammation after myocardial infarction [23]. It is conceivable that improved drainage function pauses the vicious cycle of adventitial inflammation. Abrogating inflammatory signals derived from the coronary adventitia suppressed Rho-kinase activation and resultant medial VSMC hypercontraction. Lymphatic vessels, adventitial/ PVAT inflammation, and Rho-kinase expression/activation all showed significant correlations with coronary vasoconstriction to serotonin. Taken together, cardiac lymph-angiogenesis may play a central role in the beneficial effects of the LIPUS therapy on coronary functional abnormalities after DES implantation in pigs in vivo (**Fig 6**).

## Clinical implications of the LIPUS therapy

In order to treat coronary adventitial inflammation, it is ideal to administer anti-angiogenic or anti-inflammatory agents to patients. However, systemic use of these drugs is hampered by their serious side effects [24, 25]. The anti-angiogenic strategy for vasa vasorum may cause intramural hemorrhage as a consequence of increased vascular permeability [26]. We previously demonstrated that catheter-based renal sympathetic denervation is effective to suppress DES-induced coronary hyperconstricting responses via the kidney-brain-heart-axis in pigs in vivo [6]. We also developed a balloon-catheter based technique to deliver adenovirus-mediated gene transfer of C-type natriuretic peptide gene in pigs in vivo [27]. However, these approaches required multiple invasive steps. In the present study, we were able to demonstrate that LIPUS is a novel non-invasive therapeutic option for CAD with adventitial inflammation. Since the intensity of ultrasound used in the LIPUS therapy is below the upper limit of acoustic output standards for diagnostic devices, it causes no compression, heat, or discomfort [10–14]. Thus, the LIPUS therapy could be a novel and safe therapeutic approach for CAD in clinical practice.

Although atherosclerotic pig models have been proposed to examine the mechanisms of CAD [28], it is impossible to distinguish the 'inside-out' pathway with the 'outside-in' pathway in the development of coronary lesions. Thus, it was necessary to have a non-atherosclerotic pig model in which coronary lesions can be developed by the 'outside-in' pathway. Our previous studies demonstrated that coronary adventitial inflammation can be extensively induced by a simple DES implantation technique in a non-atherosclerotic porcine coronary artery [4–9], and that vasa vasorum augmentation [4, 5], PVAT inflammation [7], autonomic nervous system [6], and lymphatic vessel dysfunction [9] are all associated with coronary hyperconstricting responses in pigs with DES implantation. Moreover, by using multimodality imaging approach with optical coherence tomography and positron emission tomography, we demonstrated that coronary adventitial/PVAT inflammatory changes are markedly increased in patients with vasospastic angina compared with control subjects [29, 30]. PVAT inflammation has also been demonstrated in human coronary atherosclerotic lesions in vivo [31]. Since coronary adventitial inflammation is implicated in the plaque disruption [2, 32], it is also conceivable that the LIPUS therapy stabilizes the culprit lesions in patients with ACS.

## Study limitations

Several limitations should be mentioned for the present study. First, although we demonstrated the preventive effect of the LIPUS therapy, it remains to be examined how long its effects last. Second, since the LAD coronary artery runs straight down on the frontside of the heart, it was easy to visually distinguish the stented coronary segments from adjacent vaoconstricting sites. Meanwhile, we did not address the effects of the LIPUS therapy on the LCx or right coronary arteries without DES implantation. Third, the detailed molecular mechanisms remains to be examined for the different effects of the LIPUS therapy in inflammatory (lymphangiogenesis) vs. ischemic tissues (angiogenesis). Fourth, we did not directly examine the effect of the LIPUS therapy on endothelial functions. However, the results with bradykinin with and without L-NMMA suggest that endothelial function was not affected by the LIPUS therapy. Fifth, although the number of lymphatic vessels was strongly associated with adventitial inflammatory changes, their causal relationships were not fully determined. Sixth, in order to avoid the effects of female hormones on coronary vasomotion [33], we used male pigs after castration. Thus, the present findings remain to be confirmed in female pigs. Seventh, the effects of the LIPUS therapy on atherosclerotic lesions remain to be examined in future studies. We have recently reported that the number and caliber of lymphatic vessels were increased at the EES stent edges in pigs [9]. The effect of LIPUS on atherosclerotic lesions or a DES with biodegradable polymers [5] remain unknown. Finally, the safety and efficacy of the LIPUS therapy should be examined in CAD patients in the clinical setting in future studies.

## Conclusions

In the present study, we were able to demonstrate that our LIPUS therapy is effective and safe for DES-induced coronary hyperconstricting responses in pigs in vivo, for which improved lymphatic drainage function for adventitial inflammatory changes may play a central role.

## Supporting information

**S1 Table. Stent implantation procedure parameters.** Results are expressed as mean ± standard error of mean (SEM). Stent diameter was calculated by averaging the diameters at the proximal edge, mid portion, and distal edge of the stented coronary artery. Overstretch ratio was calculated as the stent diameter divided by target vessel diameter. Distal overstretch ratio was calculated as the distal stent diameter divided by distal reference vessel diameter. LIPUS = low-intensity pulsed ultrasound.
(TIFF)

**S2 Table. Major resources tables.** eNOS = endothelial nitric oxide synthase; LYVE-1 = lymphatic vessel endothelial hyaluronan receptor-1; pMYPT1 = phosphorylated myosin phosphatase target subunit-1; ROCK = Rho-associated protein kinase; VEGF = vascular endothelial growth factor; VEGF-R = VEGF-receptor; vWF = von Willebrand factor.
(TIFF)

**S3 Table. Histomorphometry of the stent edges.** Results are expressed as mean±SEM. In the Masson's trichrome histology, the adventitial area was calculated by the following formula: area outside the external elastic lamina within a distance of thickness of intima plus media—vessel area. Abbreviations as in **S1 Table**.
(TIFF)

**S4 Table. Quantitative coronary angiography for vasodilating responses.** Results are expressed as mean±SEM. Coronary vasodilating responses at the stent edge segments 5 mm

apart from the stent were assessed by quantitative coronary angiography in response to nitro-glycerin (10 μg/kg, IC), bradykinin (0.1 μg/kg, IC) alone and bradykinin after L-NMMA (1 mg/kg, IC) at 1 month after stent implantation. Percent changes in diameter are expressed as those from the baseline level (contrast medium only). The mean value of vasodilating responses at the proximal and the distal stent edges are presented. L-NMMA = N$^{G}$-mono-methyl-L-arginine; other abbreviation as in **S1 Table**.
(TIFF)

**S1 Fig. Acoustic pressure of LIPUS and schematics for a LIPUS session.** (**A**) Acoustic pressure at 1 cycle which is chosen for clinically available diagnostic ultrasound devices, and (**B**) at 32 cycles for the present study. (**C**) Schematics explaining that 3 sites for one LIPUS session, including the sites proximal and distal to the stents, middle portions of the stents. Target segments were determined by co-registering radiopaque echo probe with the stent architecture. LIPUS = low-intensity pulsed ultrasound.
(TIFF)

**S2 Fig. Inter- and intra-observer variabilities for measurement of coronary diameter at stent edges after intracoronary nitroglycerin.** (**A**) Interobserver variability. (**B**) Intraobserver variability. Bland-Altman plots of differences in average coronary diameter 5 mm away from stent edges after intracoronary nitroglycerin (10 μg/kg) are shown. Green lines show 95% CI; yellow lines, mean.
(TIFF)

**S3 Fig. Immunohistochemistry for lymphatic vessels far from stent distal edges.**
(TIFF)

**S4 Fig. Immunohistochemistry for vasa vasorum and related angiogenic factors.** (**A** and **E**) Low-magnified MT stainings, and representative immunohistology of (**B** and **E**) vWF-positive vasa vasorum, (**C** and **G**) VEGF-A-positive cells, and (**D** and **H**) VEGFR2-positive cells in the adventitia of vasoconstricting portions. As denoted by arrows in (**B through D** and **F through H**), cellular structures positive for (**I**) vWF, (**J**) VEGF-A, and (**K**) VEGFR2 were significantly increased in the LIPUS group as compared with the sham group. (**L**) No correlation between vasa vasorum vs. eNOS-positive cells. Results are expressed as mean±SEM.
eNOS = endothelial nitric oxide synthase; MT = Masson's trichrome; SEM = standard error of mean; VEGF = vascular endothelial growth factor; VEGFR = vascular endothelial growth factor receptor; vWF = von Willebrand factor; other abbreviations as in **S1 Fig**.
(TIFF)

**S5 Fig. Vessel wall thickness and its relationship with eNOS expression.** (**A**) Comparison of wall thickness of lymphatic vessels vs. vasa vasorum. (**B**) A significant positive correlation between lymphatic vessel wall thickness and eNOS positive vessels. Results are expressed as mean±SEM (**A**). Abbreviations as in **S1** and **S2 Figs**.
(TIFF)

**S6 Fig. Histological analysis of adipocytes and secreted adiponectin.** (**A** and **C**) MT stainings explaining the individual adipocyte measurement, and corresponding immunohistology of (**B** and **D**) adiponectin staining-positive cells of the perivascular adipose tissue of the vaso-constricting portions. Diameters in the perpendicular maximum and minimum axes (denoted by arrows) were measured (**A** and **C**). (**E**) Adipocyte was significantly smaller in the LIPUS group as compared with the sham group. (**F**) Adiponectin % positive area was significantly greater in the LIPUS group as compared with the sham group. Results are expressed as mean

±SEM. Abbreviations as in **S1** and **S2 Figs**.
(TIFF)

**S1 Data. Data.**
(XLSX)

## Acknowledgments

The authors thank Y. Watanabe and H. Yamashita for their excellent technical assistance, and Asahi Kasei Pharma for providing hydroxyfasudil.

## Author Contributions

**Conceptualization:** Yasuharu Matsumoto, Hiroaki Shimokawa.

**Data curation:** Tasuku Watanabe, Hirokazu Amamizu, Koichi Sato, Susumu Morosawa.

**Formal analysis:** Tasuku Watanabe, Kensuke Nishimiya, Hirokazu Amamizu.

**Funding acquisition:** Yasuharu Matsumoto, Kensuke Nishimiya, Kazuma Ohyama, Hiroaki Shimokawa.

**Investigation:** Tasuku Watanabe, Hirokazu Amamizu, Jun Sugisawa, Satoshi Tsuchiya.

**Methodology:** Tomohiko Shindo, Susumu Morosawa, Kazuma Ohyama, Hiroaki Shimokawa.

**Project administration:** Yasuharu Matsumoto, Hiroaki Shimokawa.

**Supervision:** Yasuharu Matsumoto, Satoshi Yasuda, Hiroaki Shimokawa.

**Validation:** Tasuku Watanabe, Kensuke Nishimiya, Koichi Sato, Susumu Morosawa, Kazuma Ohyama, Tomomi Watanabe-Asaka, Moyuru Hayashi, Yoshiko Kawai, Jun Takahashi, Satoshi Yasuda.

**Visualization:** Tasuku Watanabe.

**Writing – original draft:** Tasuku Watanabe.

**Writing – review & editing:** Yasuharu Matsumoto, Kensuke Nishimiya, Hiroaki Shimokawa.

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
