## [Decision Letter · Decision Letter 0]

21 Jul 2021

PONE-D-21-19896

Low-intensity pulsed ultrasound therapy suppresses coronary adventitial inflammatory changes and hyperconstricting responses after coronary stent implantation in pigs in vivo

PLOS ONE

Dear Dr. Shimokawa,

Thank you for submitting your manuscript to PLOS ONE. After careful consideration, we feel that it has merit but does not fully meet PLOS ONE’s publication criteria as it currently stands. Therefore, we invite you to submit a revised version of the manuscript that addresses the points raised during the review process.

The reviewers commented favorably on your manuscript, but had some worthwhile suggestions. The authors should address the remaining issues, including the discussion of experimental limitations. I am pleased to accept your manuscript, based on your revising it.

In addition to the reviewers’ comments, the authors should address the following comments:

#1 Only male pigs were used and the effect of LIPUS on female is not shown. If available, show female data. If not, discuss its limitation.

#2 Only everolimus-eluting stent was used, but the authors use the general term “DES” in the main text. Provide rationale to use everolimus-eluting stent. Are there any differences in the effects of LIPUS therapy on stent edge vasoconstriction and eNOS activity among DESs and other types of stent?

#3 Check the first page of references section.

We look forward to receiving your revised manuscript.

Kind regards,

Michinari Nakamura, MD

Academic Editor

PLOS ONE

Journal Requirements:

"This work was supported in part by the grants-in-aid for the Scientific Research (18K15877,

19K11762, 19K17511), Mitsui Sumitomo Insurance Welfare Foundation, and Sakakibara

Memorial Research Grant from the Japan Research Promotion Society for Cardiovascular

Diseases."

"The funders had no role in study design, data collection and analysis, decision to

publish, or preparation of the manuscript."

Reviewers' comments:

Reviewer's Responses to Questions

**Comments to the Author**

1. Is the manuscript technically sound, and do the data support the conclusions?

Reviewer #1: Yes

Reviewer #2: Yes

Reviewer #3: Yes

2. Has the statistical analysis been performed appropriately and rigorously? 

Reviewer #1: Yes

Reviewer #2: Yes

Reviewer #3: Yes

3. Have the authors made all data underlying the findings in their manuscript fully available?

Reviewer #1: Yes

Reviewer #2: Yes

Reviewer #3: Yes

4. Is the manuscript presented in an intelligible fashion and written in standard English?

Reviewer #1: Yes

Reviewer #2: Yes

Reviewer #3: Yes

5. Review Comments to the Author

Reviewer #1: In this manuscript, the authors assessed the effect of low intensity pulsed ultrasound (LIPUS) on the vascular inflammation after coronary stent implantation. The authors reported that stent edge vasoconstriction was significantly suppressed in LIPUS-treated group compared to sham group. In addition, the LIPUS treatment significantly improved the lymph transport speed. These functional improvements were rationalized by the histological analysis. The reviewer thinks the manuscript was well-written, results were clearly presented and the data supports the conclusion.

Reviewer #2: The authors examined whether our LIPUS therapy suppressed coronary hyperconstricting responses in pigs after DES implantation in vivo, and they found that coronary vasoconstricting responses to serotonin in LAD at DES edges were

significantly suppressed in the LIPUS group compared to the sham group.

In addition, inflammatory changes and Rhokinase activity were significantly suppressed in the LIPUS group at the DES edges, compared to the sham group.

In the clinical settings, coronary adventitial inflammation may be associated with DES induced coronary hyperconstricting responses, which is related to refractory responses for medications.

Although data were obtained in the pig models, the findings are interesting and have an impact.

The reviewer has only a few criticisms.

Specific comments

1. Inter- and intra-observer variabilities for QCA should be added.

2. LIPUS improves cardiac lymphatic vessel function.

Does this mean that microcirculation ameliorates after the LIPUS therapy?

3. As the authors described limitation, the reviewer wonders why the authors choose stented segment in LAD for ROI. Are there any differences for responses among three coronary vessels? How do the authors think?

Reviewer #3: The authors describe a study of 16 pigs that were treated with Promus DES to LAD and randomly assigned to receive sham or LIPUS therapy (8 in each group). The pigs receiving LIPUS therapy had less adventitial inflammation, faster lymphatic transport in the adventitial tissue and less vasoconstriction to serotonin.

Authors propose the following pathologic mechanism of coronary hyperconstiction after DES placement:

1. Stent placement induces inflammation which leads to hyperconstricting coronary response.

2. LIPUS is hypothesized to change shear stress in lymphatic blood vessels which then release eNOS.

3. LIPUS (via eNOS) decreases inflammation (The authors demonstrate that there were fewer CD68+ macrophages and less IL-1beta expression in the adventitias of LIPUS treated blood vessels, but more expression of the anti-inflammatory adiponectin.)

4. LIPUS (via eNOS) stimulates lymphangiogenesis (The authors show increased expression of factors that stimulate lymphangiogenesis (LYVE-1, VEGF-C, VEGFR3) but not angiogenesis (VEGF-A, VEGFR2) in the LIPUS treated group.)

5. More lymphatic vessels means faster lymphatic transport and less inflammation (The number of lymphatic vessels correlated inversely with the presence of CD68+ macrophages and the expression of IL-1beta.).

6. Both less inflammation (Fig 5I, J) and faster lymphatic transport (Fig 3H) correlated with less vasoconstriction with serotonin.

7. Vasoconstriction with serotonin appears to be mediated by Rho-kinase (which, in turn, is expressed more in inflammation) because administration of hydroxyfusadil, a Rho-kinase inhibitor, leads to disappearance of vasoconstriction. Rho-kinase expression was decreased in the LIPUS group as compared to sham group.

The paper is extremely well written and concise, it relays the key message perfectly and needs little or no editing prior to publication.

Minor comments:

1. Avoid use of abbreviations unless absolutely necessary. For instance, page 3, line 20, you use CAG for coronary angiography which had not previously been defined and is likely not a necessary abbreviation at all.

2. Did the sham group also receive anesthesia for each sham LIPUS session?

3. How precise do you think LIPUS treatment was geographically with respect to beam width relative to target size? Are there any “off target” effects of LIPUS?

4. The vessels that were stented were normal, i.e. without large atherosclerotic plaque burden which limits the generalizability of the study findings to clinical settings. The authors did explain their choice of the non-atherosclerotic vessel in the discussion section but should consider acknowledging this limitation in the limitations section as well.

5. The authors note that they did not examine in detail the effects of LIPUS therapy on endothelial function which is known to also be disrupted at the stent edges. You used bradykinin and showed normal vasodilatory response in both LIPUS and sham groups. Why did you choose bradykinin over acetylcholine?

6. How is hypercontractility related to restenosis at the stent edges?

6. PLOS authors have the option to publish the peer review history of their article (what does this mean?). If published, this will include your full peer review and any attached files.

Reviewer #1: No

Reviewer #2: No

Reviewer #3: **Yes: **Natalija Odanovic

---

## [Author Response · Author response to Decision Letter 0]

19 Aug 2021

We greatly appreciate the Editor and Reviewers for the favorable comments on our work.

We responded to specific comments in each "R-1 Responses" file.

---

## [Decision Letter · Decision Letter 1]

25 Aug 2021

Low-intensity pulsed ultrasound therapy suppresses coronary adventitial inflammatory changes and hyperconstricting responses after coronary stent implantation in pigs in vivo

PONE-D-21-19896R1

Dear Dr. Shimokawa,

We’re pleased to inform you that your manuscript has been judged scientifically suitable for publication and will be formally accepted for publication once it meets all outstanding technical requirements.

Kind regards,

Michinari Nakamura, MD

Academic Editor

PLOS ONE

Additional Editor Comments (optional):

Thank you for your great work!

Reviewers' comments:

Reviewer's Responses to Questions

**Comments to the Author**

1. If the authors have adequately addressed your comments raised in a previous round of review and you feel that this manuscript is now acceptable for publication, you may indicate that here to bypass the “Comments to the Author” section, enter your conflict of interest statement in the “Confidential to Editor” section, and submit your "Accept" recommendation.

Reviewer #1: All comments have been addressed

Reviewer #2: All comments have been addressed

Reviewer #3: All comments have been addressed

2. Is the manuscript technically sound, and do the data support the conclusions?

Reviewer #1: Yes

Reviewer #2: Yes

Reviewer #3: Yes

3. Has the statistical analysis been performed appropriately and rigorously? 

Reviewer #1: Yes

Reviewer #2: Yes

Reviewer #3: Yes

4. Have the authors made all data underlying the findings in their manuscript fully available?

Reviewer #1: Yes

Reviewer #2: Yes

Reviewer #3: Yes

5. Is the manuscript presented in an intelligible fashion and written in standard English?

Reviewer #1: Yes

Reviewer #2: Yes

Reviewer #3: Yes

6. Review Comments to the Author

Reviewer #1: The reviewer feels the manuscript has been improved and now became more suitable for publication. No further requests or comments.

Reviewer #2: (No Response)

Reviewer #3: (No Response)

7. PLOS authors have the option to publish the peer review history of their article (what does this mean?). If published, this will include your full peer review and any attached files.

Reviewer #1: No

Reviewer #2: **Yes: **Hideki Ishii

Reviewer #3: **Yes: **Natalija Odanovic

---

## [Editor Report · Acceptance letter]

3 Sep 2021

PONE-D-21-19896R1 

Low-intensity pulsed ultrasound therapy suppresses coronary adventitial inflammatory changes and hyperconstricting responses after coronary stent implantation in pigs in vivo 

Dear Dr. Shimokawa:

I'm pleased to inform you that your manuscript has been deemed suitable for publication in PLOS ONE. Congratulations! Your manuscript is now with our production department. 

Kind regards, 

on behalf of

Dr. Michinari Nakamura 

Academic Editor

PLOS ONE